# DiCoH: Rethinking Self-Supervised Pretraining for Semantic Segmentation in Homogenous Medical Domains

## Abstract

Self-supervised learning (SSL) for pretraining has become critical for improving segmentation performance when labeled data is scarce. However, existing contrastive methods are primarily designed for diverse, object-centric natural images and struggle to generalize to *homogenous* medical datasets that exhibit low semantic variation across both images and pixels. Low semantic variations make aligning positive pixel-to-pixel pairs trivial and make identifying true negative pairs extremely challenging. Additionally, we identify that *architectural asymmetry*, demonstrated to stabilize contrastive pretraining, is detrimental when applied to homogeneous data. To tackle these limitations, we present **Di**verse **Co**ntrastive Learning for **H**omogeneous Data (DiCoH), an SSL pretraining framework for homogeneous medical data. DiCoH improves representation learning by diversifying positive pixel-to-pixel alignments and guaranteeing true negative pairs through a novel *hard* pixel-to-image selection strategy. Comprehensive evaluations on five medical segmentation datasets demonstrate that DiCoH significantly and consistently outranks state-of-the-art SSL methods, achieving +2.00% mIoU gains under extremely low-data conditions.[1]

## 1 Introduction

Semantic segmentation is crucial in medical image analysis, yet its success still relies on substantial pixel-wise annotated data (Li et al., 2025). Here, annotations are costly and time-consuming, often requiring expert supervision (Zhang et al., 2024). This has driven the adoption of contrastive self-supervised learning (SSL) for pretraining to reduce the need for extensive labeled data during finetuning (Kang et al., 2023; Van-Berlo et al., 2024). Contrastive SSL methods like SimCLR (Chen et al., 2020) and MoCo (He et al., 2020) achieve strong results on natural images by enforcing agreement between

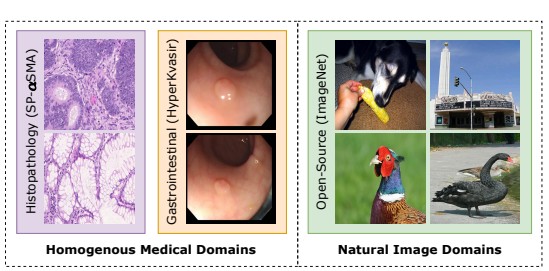

Figure 1: Homogenous medical datasets feature low inter-image diversity and lack clear semantic/object boundaries in contrast to open-source data.

augmented *views* of the same image while contrasting them with views from different images. However, their reliance on image-level objectives has proven suboptimal for dense prediction tasks like segmentation (Xie et al., 2021; Shen et al., 2023).

As a result, recent work has shifted toward multi-level contrastive objectives at the pixel (e.g., PixPro (Xie et al., 2021), CP2 (Wang et al., 2022a)), region (ConCL (Yang et al., 2022)), and cluster levels (CA$^2$CL (Li et al., 2025)). These methods propose and establish several key refinements as standard practice: **(1)** Define positive pixel-to-pixel pairs across views either by spatial proximity (Xie et al., 2021; Wang et al., 2022a) or similarity scores (Wang et al., 2021); **(2)** Applying hard negative mining strategies to improve feature separation by selectively pushing apart the most challenging

---

[1]Code will be released on acceptance

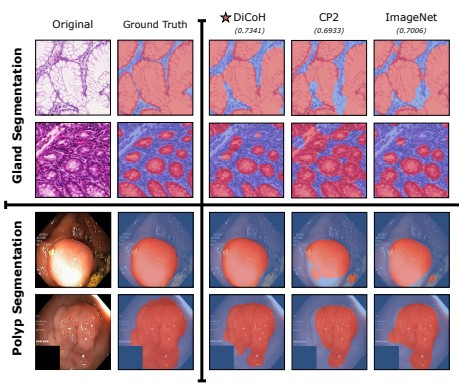 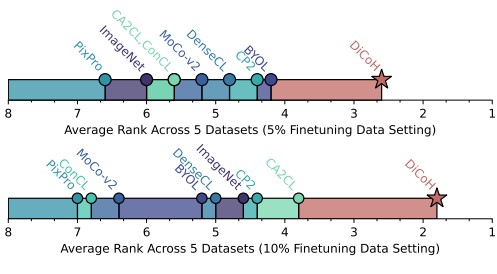

(a) Impact of SSL pretraining on segmentation quality.     (b) Average ranks of pretrained networks.

Figure 2: **Impact of SSL pretraining on homogeneous medical segmentation tasks.** **(a)** Example results for gland (GlaS (Sirinukunwattana et al., 2017)) and polyp (Kvasir (Borgli et al., 2020)) segmentation with only 5% labeled data. **(b)** Average ranks of networks pretrained with state-of-the-art SSL methods across five medical benchmarks under 5% and 10% labeled data.

pixel or image-level pairs (Ash et al., 2022; Zhang et al., 2023); and **(3)** Using asymmetrical siamese architectures to stabilize and improve pretraining (Chen & He, 2021; Wang et al., 2022b; Grill et al., 2020) with separate projection networks for pixel and image-level features (DenseCL (Wang et al., 2021), CP2 (Wang et al., 2022a)).

Despite these advances, most refinements are tuned for diverse, object-centric natural images (e.g. ImageNet (Deng et al., 2009), COCO (Caesar et al., 2018)) and do not transfer well to *homogeneous* medical datasets, which exhibit low inter-image diversity and ambiguious object boundaries (Fig. 1). In such data, conventional strategies face three challenges:

i **Trivial positive pairs**: In homogeneous domains, many pixels exhibit similar or nearly identical textures and semantics. As a result, aligning "positive pairs" (the same pixel across two augmentations) becomes trivial, providing little incentive for the network to learn discriminative representations.

ii **False negative pairs**: Hard negative mining is designed to improve feature separation by pushing apart the most confusing examples. However, in homogeneous data, many pixels are semantically aligned yet mistakenly selected as negatives, a phenomenon known as *semantic collisions* (Zhang et al., 2023; Ash et al., 2022). This forces the network to separate features that actually belong together.

iii **Feature variance collapse**: Asymmetric siamese architectures are often used to stabilize training by suppressing variance in the target branch (Grill et al., 2020; Chen & He, 2021). While effective on natural images, in homogeneous domains this variance reduction collapses meaningful feature differences, making it harder to separate positives from negatives and exacerbating the above issues.

Addressing these challenges is importance since SSL for medical imaging is a growing an high-impact task. Domain-specific approaches such as ConCL (Yang et al., 2022) and CA2CL (Li et al., 2025) address aspects of pathology data but lack comprehensive benchmarks across multiple homogeneous medical domains under ultra–low-data conditions. Moreover, many studies fail to compare against the widely adopted ImageNet-only pretraining standard (Sanderson & Matuszewski, 2024; Xie et al., 2019; Menegola et al., 2017; VanBerlo et al., 2024), leaving open questions about how medical SSL should be evaluated in practice.

To address these gaps, we propose **Di**verse **Co**ntrastive Learning for **H**omogeneous Data (DiCoH), an SSL pretraining framework tailored for homogeneous medical domains. DiCoH introduces three key innovations: **(i)** diversified one-to-many positive pixel alignments via spatial and similarity maps, **(ii)** robust pixel-to-image negatives that reduce semantic collisions, and **(iii)** a symmetric siamese architecture that preserves feature variance. Through extensive evaluations on five medical

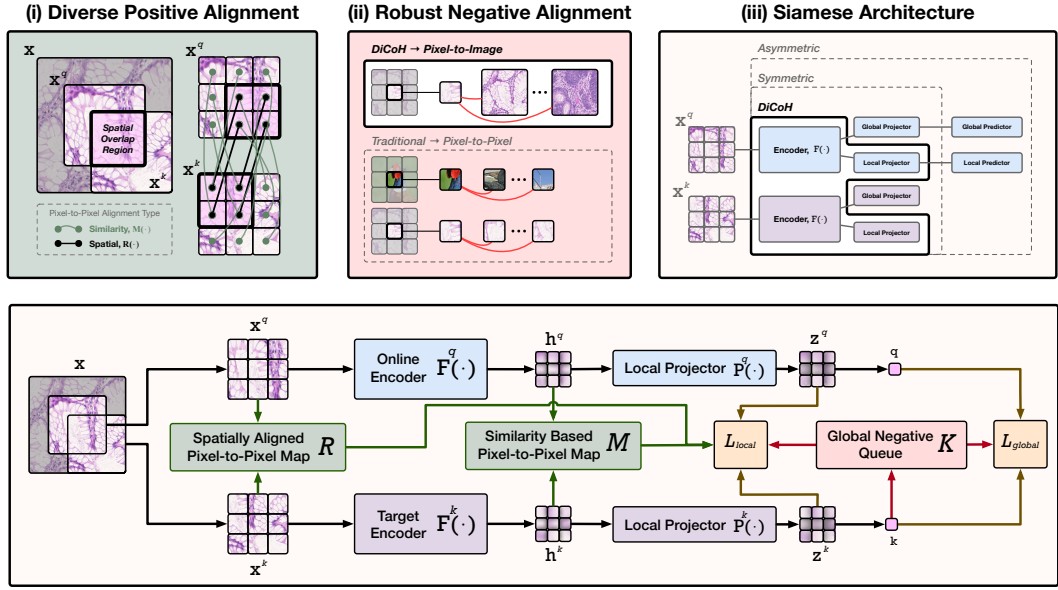

Figure 3: **System overview of DiCoH.** An input image ($\mathbf{x}$) is augmented into query ($\mathbf{x}^q$) and key ($\mathbf{x}^k$) views, which are then processed by symmetric online and target encoders/projectors. DiCoH introduces three components to address the challenges of homogeneous medical datasets: (i) **Diverse Positive Alignment** combines spatial (same pixel location) $R$ and similarity-based (nearest neighbor across views) $M$ maps to create pixel-to-pixel positive pairs; (ii) **Robust Negative Alignment** replaces conventional *pixel-to-pixel* negatives with *pixel-to-image* negatives drawn from a global (image-level representations) queue, $\mathcal{K}$, explicitly reducing semantic collisions; and (iii) a **Symmetric Siamese Architecture** prevents feature variance collapse seen in conventional asymmetric designs. Together, these yield more discriminative representations from homogenous data.

segmentation benchmarks under 5%–10% labeling regimes, we show that DiCoH consistently achieves the top average rank, delivering up to +2.0% mIoU gains over state-of-the-art baselines; refer to Fig. 2.

We summarize our contributions as follows:

1. **Address SSL limitations:** We analyze why trivial positives, false negatives, and asymmetric designs degrade contrastive pretraining in homogeneous medical data.

2. **DiCoH framework:** We introduce diversified positives (spatial + similarity), pixel-to-image negatives, and a symmetric siamese design to address these challenges.

3. **Comprehensive benchmarks:** We provide the first set of comprehensive benchmarks of SSL pretraining methods for semantic segmentation in *homogeneous* medical domains under extreme finetuning label scarcity using the ImageNet-initialized pretraining standard.

## 2 RELATED WORKS

**Global consistency frameworks.** The *cross-view consistency* learning task emerged as a powerful way to learn transferable representations in a self-supervised manner. This task seeks to align representations of two augmentations (*views*) of the same image while contrasting those of different images in the dataset. Foundational approaches like SimCLR (Chen et al., 2020) and MoCo-v2 (He et al., 2020), built on top of the ubiquitous InfoNCE (Oord et al., 2018) loss, are the foundation of most contrastive frameworks to date, supervised (Khosla et al., 2020; Wei et al., 2023; Gupta et al., 2023; Yao et al., 2022; Cai et al., 2024) or self-supervised (Wang et al., 2023b; Wu et al., 2023; Dai et al., 2024; Lin et al., 2022; Jenni et al., 2023). Surprisingly, widely influential works like BYOL (Grill et al., 2020) and SimSiam (Chen & He, 2021) only align views without needing

the contrastive component. Consequently, SSL learning strategies like DINO (Caron et al., 2021), BarlowTwins (Zbontar et al., 2021), VICReg (Bardes et al., 2021) adopt this *non-contrastive* approach in favour of its simplicity. These global consistency frameworks deliver strong results for general vision tasks, particularly image classification. However, they struggle in applications requiring detailed pixel-level predictions, motivating the shift towards local consistency framework.

**Local consistency frameworks.** However, recent literature argues that image-level consistency tasks are sub-optimal for dense prediction tasks like semantic segmentation (Shen et al., 2023; Xie et al., 2021; Lebailly et al., 2024). Consequently, newer works extend global *cross-view consistency* learning to a local level. These methods can be categorized by the granularity at which consistency is enforced. Methods like DenseCL (Wang et al., 2021) and CP2 (Wang et al., 2022a) enforce consistency on a pixel level; the former aligns pixels one-to-one based on similarity scores, while the latter takes a one-to-all approach. Methods like PixPro (Xie et al., 2021), CrOC (Stegmüller et al., 2023), CrIBo (Lebailly et al., 2024), and DetCon (Hénaff et al., 2021) seek to enforce consistency between groups of pixels they identify with being semantically aligned through a combination of heuristics and various nearest neighbour retrieval strategies. *Cross-view consistency* frameworks have primarily been developed for diverse object-centric datasets like ImageNet (Deng et al., 2009) and COCO-Stuff (Caesar et al., 2018). However, a lack of work comprehensively studies them on low-diversity, homogenous data typically found in medical, manufacturing and agricultural domains. This lack of targeted research in low-diversity datasets forces a more tailored approach.

**SSL pretraining in medical domains.** Due to the lack of large annotated datasets, SSL created substantial interest in medical imaging tasks in domains such as gastrointestinal endoscopy, histopathology, cardiology and neurology (Sanderson & Matuszewski, 2024; Kang et al., 2023; Huang et al., 2023; Shurrab & Duwairi, 2022; Varoquaux & Cheplygina, 2022; VanBerlo et al., 2024). Foundational *cross-view consistency* methods, such as MoCo-v2 (He et al., 2020) and Sim-CLR (Chen et al., 2020), have proven to be highly effective in these domains, often matching or surpassing the performance of supervised learning approaches (Wang et al., 2023a). These methods are usually employed with domain-specific adjustments include data augmentation strategies (Kang et al., 2023; Alomar et al., 2023; Chen & Lu, 2023; Kebaili et al., 2023; Su et al., 2023), pretraining workflows (Azizi et al., 2023; Huang et al., 2023), uses of auxiliary medical data (Haghighi et al., 2021), and clustering strategies (Yang et al., 2022; Li et al., 2025). We build on these works with a method tailored for homogeneous medical datasets across multiple domains.

**ImageNet-centric workflows in medical domains.** Most medical studies do not compare their methods against the widely used ImageNet-only (Deng et al., 2009) pre-trained weights (VanBerlo et al., 2024; Azizi et al., 2021; Ma et al., 2022). This is significant because several studies advocate for ImageNet-only initialization as being more effective than domain-aligned SSL from scratch, followed by finetuning on medical data (Sanderson & Matuszewski, 2024; Haghighi et al., 2020; Xie et al., 2019; Menegola et al., 2017). Consequently, we evaluate SSL methods on top of ImageNet-initialized backbones.

**Histopathology-specific contrastive learning.** Despite the extensive research in natural images framework for dense tasks the benchmarks in pathology domain are quite sparse. To address this we evaluate our method against the two most recent methods tailored for contrastive learning for histopathology: ConCL (Concept Contrastive Learning)(Yang et al., 2022) and CA$^2$CL(Cluster-Aware Adversarial Contrastive Learning)(Li et al., 2025). ConCL contrasts local *concept* regions rather than whole images. CA$^2$CL introduces a cluster-aware loss to mitigate false negatives and uses adversarial augmentation to create more challenging positive pairs. Despite these advances, these methods do not generalize to other medical tasks.

## 3 METHODOLOGY

DiCoH (**Di**verse **Co**ntrastive Learning for **H**omogeneous Data) is designed to address three fundamental challenges of homogeneous datasets: (i) Trivial positive pairs that provide weak supervision, (ii) False negatives due to semantic collisions in pixel-to-pixel contrast, and (iii) Feature variance collapse caused by asymmetric siamese architectures. Fig. 3 illustrates the overall pipeline and highlights the three key contributions. The overall learning objective is formulated as:

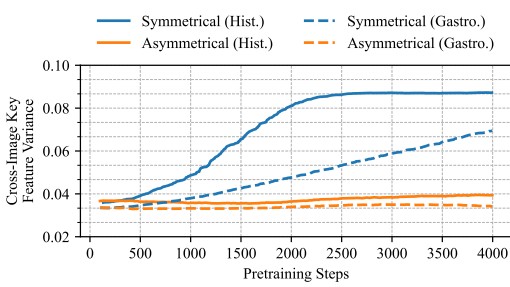 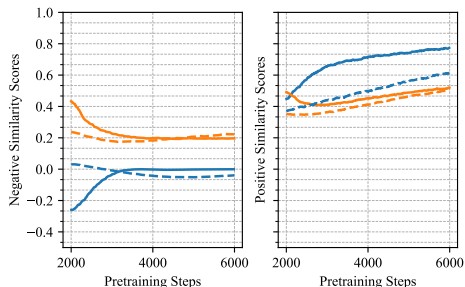

(a) Target branch key feature variances

(b) Positive/negative similarity scores

Figure 4: **Impact of architectural asymmetry on contrastive pretraining in homogeneous medical data.** (a) Asymmetric architectures reduce the variance of target branch features (Chen & He, 2021; Wang et al., 2022b) during pretraining on both gastrointestinal and histopathology datasets; **orange** < **blue**. (b) This suppression in variance leads to lower similarity scores for positive pixel pairs and higher scores for negative pairs, degrading the effectiveness of the local contrastive objective. These findings support the use of a symmetric architecture in DiCoH to preserve critical feature differences and improve contrastive alignment, as discussed in Sec. 3.4.

$$\mathcal{L} = \lambda\mathcal{L}_{local} + (1 - \lambda)\mathcal{L}_{global}, \tag{1}$$

$\mathcal{L}_{local}$ learns local (pixel-level) representations; $\mathcal{L}_{global}$ learns global (image-level) context.

## 3.1 PRELIMINARIES

**Network.** DiCoH follows a siamese (Chen & He, 2021) architecture where an input image $\mathbf{x} \in \mathbb{R}^{C \times H \times W}$ undergoes two augmentations $t_1, t_2$ to generate a query *view* $\mathbf{x}^q = t_1(\mathbf{x})$ and a key *view* $\mathbf{x}^k = t_2(\mathbf{x})$. The key view $\mathbf{x}_q \in \mathbb{R}^{C \times H \times W}$ is passed through an *online* encoder $F_q(\cdot)$ and projector $P_q(\cdot)$ network to compute local (pixel-level) representations, $\mathbf{h}^q = F_q(\mathbf{x}^q) \in \mathbb{R}^{C_1 \times S^2}$ and $\mathbf{z}^q = P_q(\mathbf{h}^q) \in \mathbb{R}^{C_2 \times S^2}$. A parallel *target* network computes local (pixel-level) representations of the key view, $\mathbf{h}^k = F_k(\mathbf{x}^k)$ and $\mathbf{z}^k = P_k(\mathbf{h}^k)$. The *target* newtork parameters are updated via exponential moving average of the *online* network.

**Global Loss.** Contrastive pretraining at the image (global) level, while not ideal as a sole objective (Xie et al., 2021; Shen et al., 2023), greatly benefits dense prediction tasks like semantic segmentation. Typical SSL pretraining methods use separate projectors to capture global and local features (Wang et al., 2021; Xie et al., 2021). DiCoH, however, derives global key and query representations $\mathbf{q}, \mathbf{k} \in \mathbb{R}^{C_2 \times 1}$ directly from their respective local representations $\mathbf{z}^q, \mathbf{z}^k \in \mathbb{R}^{C_2 \times S^2}$ via normalize average pooling. This ensures that $\mathcal{L}_{global}$ optimizes the underlying local representations and enables the pixel-to-image negative sampling strategy introduced in Sec. 3.3. The InfoNCE loss function (Oord et al., 2018) is employed to pull $\mathbf{q}$ close to $\mathbf{k}$ while pushing it away from other keys in a negative queue $\mathcal{K}$:

$$\mathcal{L}_{global}(\mathbf{q}, \mathbf{k}) = -\log \frac{\exp(\mathbf{q} \cdot \mathbf{k}/\tau)}{\sum_{\mathbf{c} \in \{k\} \cup \mathcal{K}} \exp(\mathbf{q} \cdot \mathbf{c}/\tau)}. \tag{2}$$

**Local Loss.** The local contrastive loss $\mathcal{L}_{local}$ enforces pixel-wise feature consistency by aligning positive pixel pairs while separating negative ones. Given the pixel-level (local) representations of our query and key images, $\mathbf{z}^q, \mathbf{z}^k \in \mathbb{R}^{C_2 \times S^2}$ positive pairs are sampled using spatial and similarity pixel-to-pixel maps (Sec. 3.2). Negative pairs are selected according to our robust pixel-to-image strategy (Sec. 3.3). The local contrastive objective also follows an InfoNCE formulation sharing the same negative queue $\mathcal{K}$ as $\mathcal{L}_{global}$.

## 3.2 DIVERSIFYING POSITIVE PIXEL-TO-PIXEL ALIGNMENT

In contrastive SSL, the goal of positive alignment is to encourage the network to recognize semantically corresponding pixels across augmented views $(\mathbf{x}^q, \mathbf{x}^k)$. However, in homogeneous medical datasets, where many pixels share nearly identical textures (e.g., repeated glandular tissue or uniform background), spatially aligned pixels across augmentations are almost always trivially similar. This triviality limits the discriminative power of pretraining because the network can minimize the objective without learning meaningful distinctions between subtle but clinically important structures.

In response, DiCoH diversifies positive alignment by combining two strategies (Fig. 3):

1. A **Spatial Alignment Map** ($R$) pairs each query pixel with its corresponding key pixel at the same spatial location. This provides reliable supervision, as these pixels are guaranteed to be semantically aligned across augmentations since they come from the same place in the original image $\mathbf{x}$. Formally, for a pixel-level query representation vector $\mathbf{z}_i^q$, the map $R$ selects its aligned key pixel $\mathbf{z}_j^k$, where $j = R(i)$.

2. A **Similarity Alignment Map** ($M$) expands supervision beyond the overlapping regions of the two views (shown in Fig. 3). This one-to-many strategy is particularly valuable in homogeneous data: pixels in different regions often represent the same tissue type or structure, and matching them exposes the model to a richer set of positive pairs. Formally, for a pixel-level query representation vector $\mathbf{z}_i^q$, the map $M$ selects its most similar key pixel $\mathbf{z}_j^k$, where $j = M(i) = \arg\max_j \ sim(\mathbf{h}_i^q, \mathbf{h}_j^k)$ and $sim(\cdot, \cdot)$ represents cosine similarity.

By combining $R$ and $M$, DiCoH prevents the pretraining task from collapsing into trivial alignment and instead supplies diverse, semantically valid correspondences. This encourages the model to learn more discriminative local (pixel-level) features, even when visual diversity is low.

## 3.3 ROBUST PIXEL-TO-IMAGE NEGATIVES

In contrastive SSL, the role of negative pairs is to push apart features that should be distinct, ensuring that representations capture discriminative structure. However, in homogeneous medical datasets pixels are visually and semantically similar and conventional pixel-to-pixel negatives frequently misclassify aligned pixels as negatives. These **semantic collisions** (Zhang et al., 2023; Ash et al., 2022) are problematic since they force the network to separate features that actually belong together.

To reduce collisions, DiCoH replaces pixel-to-pixel negatives with **pixel-to-image negatives**: each query pixel $\mathbf{z}_i^q$ is contrasted not against other individual pixels, but against global image embeddings $\mathbf{k} \in \mathcal{K}$ (introduced in Sec. 3.1) drawn from a memory queue $\mathcal{K}$ of other images (Fig. 3). These image-level (global) vectors, are semantically broader and less likely to overlap with the query pixel's content, making them safer surrogates for negatives in low-diversity settings.

We further improve discrimination through **hard negative mining**: among all pixel-to-image pairs, we select the upper quartile (75th percentile) of most similar pairs as negatives. This ensures the model learns to separate subtle but important differences, while avoiding unstable training caused by overly aggressive sampling. We compare different thresholding quartiles in Table 4d and confirms that the 75th percentile (harder negatives) yields the best performance.

Given the local representations of our query and key images, $\mathbf{z}^q, \mathbf{z}^k$, and selected negative samples, we can now formulate our local loss:

$$\mathcal{L}_{local}(\mathbf{z}^q, \mathbf{z}^k) = \mathcal{L}_{spatial} + \mathcal{L}_{similar}, \tag{3}$$

where the spatial loss is defined as:

$$\mathcal{L}_{spatial} = \frac{-1}{|R|} \sum_i^{|R|} \log \frac{\exp(\mathbf{z}_i^q \cdot \mathbf{z}_{R(i)}^k / \tau)}{\sum_{\mathbf{z}^c \in \{\mathbf{z}_{R(i)}^k\} \cup \mathcal{N}} \exp(\mathbf{z}_i^q \cdot \mathbf{z}^c / \tau)} \tag{4}$$

and the similarity-based loss is defined as:

$$\mathcal{L}_{similar} = \frac{-1}{|M|} \sum_{i}^{|M|} \log \frac{\exp(\mathbf{z}_i^q \cdot \mathbf{z}_{M(i)}^k / \tau)}{\sum_{\mathbf{z}^c \in \{\mathbf{z}_{M(i)}^k\} \cup \mathcal{N}} \exp(\mathbf{z}_i^q \cdot \mathbf{z}^c / \tau)} \tag{5}$$

$\mathcal{N}$ is the subset of hard negative samples from $\mathcal{K}$, a fixed length queue of image-level $k$ vectors.

### 3.4 Importance of Architectural Symmetry

Modern SSL frameworks commonly use an asymmetric siamese architecture (Fig. 3) to stabilize pretraining by reducing the variance of the target branch features (Wang et al., 2022b; Chen & He, 2021; Cai et al., 2021; He et al., 2020; Grill et al., 2020). This is done by concatenating a projection network on the target branch. Consistent with prior research, we observe in Fig. 4a that target feature variance drops when using an asymmetric architecture. This drop in variance correlates with lower pixel-level key-query positive similarity scores and higher negative similarity scores; refer to Fig.4b, making the local contrastive objective $\mathcal{L}_{local}$ harder to minimize. This results in significantly poorer segmentation performance across all gastrointestinal and histopathology datasets; refer to the ablations in Sec.4.2. Therefore, DiCoH employs a symmetric architecture.

## 4 Experiments

**Gastrointestinal Polyp Segmentation Dataset.** We assess the impact of pretraining by evaluating downstream segmentation performance on four small gastrointestinal polyp segmentation datasets, each containing approximately 600 images: Kvasir-SEG (Borgli et al., 2020), ClinicDB (Bernal et al., 2015), ColonDB (Bernal et al., 2012), and ETIS-Larib (Silva et al., 2014). These datasets were used for finetuning. For pretraining, we utilized the large, 100,000 image gastrointestinal HyperKvasir dataset (Borgli et al., 2020).

**Histopathology Gland Segmentation Dataset.** We evaluate the impact of pretraining using the popular MICCAI 2015 Gland Segmentation (GlaS) dataset (Sirinukunwattana et al., 2017), which contains approximately 200 images, for finetuning. For this dataset, pretraining is performed using the histopathology dataset SP-$\alpha$SMA (Komura, 2022), containing around 40,000 images.

**Pretrain-Finetune Protocol.** All methods are first initialized with ImageNet-supervised (Deng et al., 2009) pretrained weights, as is common in industry (Sanderson & Matuszewski, 2024; Xie et al., 2019; Menegola et al., 2017; VanBerlo et al., 2024), before further pretraining on medical data. Then finetuning and evaluation was performed on the respective segmentation dataset.

**Evaluation Metrics.** To evaluate binary segmentation performance, we report the Jaccard Index (i.e., mIoU), following standard practice (Bertels et al., 2019) averaged over three Monte Carlo runs (i.e., using three different seeds) per dataset. For each finetuning dataset, models are ranked from best to worst based on their mIoU. Ranking provides a fairer way to compare methods across multiple datasets because it is robust against variations in dataset difficulty (Demšar, 2006).

**Architecture.** We used the common ResNet-50 (He et al., 2016) backbone with a DeepLabv3 (Chen, 2017) Atrous Spatial Pyramid Pooling (ASPP) segmentation head. CNN architectures remain popular for medical imaging tasks since they take less data to train (Lu et al., 2022), unlike ViTs.

**Training.** For each dataset, pretraining was conducted for 10 epochs (after ImageNet pretraining) with a total batch size of 128, ensuring that each SSL method reached saturation. An SGD optimizer (Loshchilov, 2017) was employed with a learning rate of $1 \times 10^{-3}$, weight decay of $1 \times 10^{-4}$, and images were resized to 224x224. Random augmentations were applied; random flipping, random cropping, color jitter (i.e., brightness, contrast, hue, saturation), blurring, and Gaussian noise consistent with common pretraining methods (He et al., 2020; Grill et al., 2020; Xie et al., 2021). Complete finetuning, including the backbone, for each dataset, was performed for 100 epochs with a batch size of 32 across two NVIDIA RTX A6000 GPUs. This phase consistently achieved loss saturation for each method. For finetuning, an AdamW optimizer (Loshchilov, 2017) was used with a learning rate of $1 \times 10^{-4}$, a weight decay of $1 \times 10^{-4}$ and images were resized to 352x352.

Table 1: **Evaluating SSL pertaining methods on gastrointestinal (polyp segmentation) and histopathology (gland segmentation) datasets using 5% of the finetuning data.** Note that **bold** numbers denotes best; underline, second.

| Method | Clinic | | Colon | | Etis | | Kvasir | | GlaS | | Average | |
|---|---|---|---|---|---|---|---|---|---|---|---|---|
| | Rank ↓ | mIoU ↑ | Rank ↓ | mIoU ↑ | Rank ↓ | mIoU ↑ | Rank ↓ | mIoU ↑ | Rank ↓ | mIoU ↑ | Rank ↓ | mIoU ↑ |
| ImageNet | 5 | 0.6676 | 5 | 0.5400 | 8 | 0.3043 | 5 | 0.7392 | 7 | 0.6619 | 6.0 | 0.5826 |
| BYOL | 1 | **0.6902** | 6 | 0.5373 | 9 | 0.3009 | 2 | 0.7469 | 3 | 0.7003 | 4.2 | 0.5951 |
| MoCo-v2 | 2 | 0.6831 | 9 | 0.4836 | 7 | 0.3207 | 4 | 0.7419 | 4 | 0.6973 | 5.2 | 0.5853 |
| DenseCL | 6 | 0.6673 | 1 | **0.6331** | 6 | 0.3238 | 6 | 0.7320 | 5 | 0.6953 | 4.8 | 0.6103 |
| PixPro | 9 | 0.6316 | 3 | 0.5974 | 4 | 0.3706 | 9 | 0.6978 | 8 | 0.6497 | 6.6 | 0.5894 |
| CP2 | 8 | 0.6524 | 2 | 0.6066 | 2 | 0.3918 | 1 | **0.7491** | 9 | 0.6374 | 4.4 | 0.6075 |
| ConCL | 3 | 0.6752 | 7 | 0.5205 | 5 | 0.3304 | 7 | 0.7231 | 6 | 0.6773 | 5.6 | 0.5853 |
| CA²CL | 7 | 0.6654 | 8 | 0.5135 | 3 | 0.3893 | 8 | 0.7215 | 2 | 0.7032 | 5.6 | 0.5986 |
| DiCoH | 4 | 0.6702 | 4 | 0.5603 | 1 | **0.4526** | 3 | 0.7458 | 1 | **0.7224** | **2.6** | **0.6303** |

Table 2: **Evaluating SSL pertaining methods on gastrointestinal (polyp segmentation) and histopathology (gland segmentation) datasets using 10% of the finetuning data.** Note that **bold** numbers denotes best; underline, second.

| Method | Clinic | | Colon | | Etis | | Kvasir | | GlaS | | Average | |
|---|---|---|---|---|---|---|---|---|---|---|---|---|
| | Rank ↓ | mIoU ↑ | Rank ↓ | mIoU ↑ | Rank ↓ | mIoU ↑ | Rank ↓ | mIoU ↑ | Rank ↓ | mIoU ↑ | Rank ↓ | mIoU ↑ |
| ImageNet | 6 | 0.7246 | 4 | 0.6660 | 5 | 0.4578 | 3 | 0.7878 | 5 | 0.7400 | 4.6 | 0.6752 |
| BYOL | 5 | 0.7284 | 9 | 0.6200 | 2 | 0.4864 | 2 | 0.7907 | 8 | 0.7244 | 5.2 | 0.6700 |
| MoCo-v2 | 9 | 0.6860 | 2 | 0.6756 | 9 | 0.3484 | 6 | 0.7824 | 6 | 0.7279 | 6.4 | 0.6440 |
| DenseCL | 4 | 0.7306 | 6 | 0.6562 | 7 | 0.4411 | 7 | 0.7815 | 1 | **0.7594** | 5 | 0.6738 |
| PixPro | 8 | 0.7024 | 7 | 0.6520 | 4 | 0.4683 | 9 | 0.7619 | 7 | 0.7275 | 7 | 0.6624 |
| CP2 | 2 | 0.7369 | 3 | 0.6734 | 8 | 0.4090 | 5 | 0.7833 | 4 | 0.7460 | 4.4 | 0.6697 |
| ConCL | 3 | 0.7364 | 8 | 0.6275 | 6 | 0.4548 | 8 | 0.7735 | 9 | 0.7054 | 6.8 | 0.6595 |
| CA²CL | 7 | 0.7209 | 5 | 0.6637 | 1 | **0.5347** | 4 | 0.7838 | 2 | 0.7567 | 3.8 | **0.6920** |
| DiCoH | 1 | **0.7531** | 1 | **0.6761** | 3 | 0.4794 | 1 | **0.7945** | 3 | 0.7524 | **1.8** | 0.6911 |

## 4.1 COMPARISON WITH THE STATE-OF-THE-ART

We compare DiCoH with state-of-the-art SSL pretraining methods, across multiple medical segmentation tasks with varying percentages of finetuning data. Note that the `ImageNet` baseline is simply a model initialized with ImageNet-supervised pretraining only. Table 1 and Table 2 compare SSL pretrained weights when only 5% or 10% of each dataset is available for finetuning. DiCoH outperforms prior

Table 3: Evaluating against histopathology-tailored methods on the GlaS dataset.

| | Finetuning Data Amount | | | | |
|---|---|---|---|---|---|
| Method | 5% | 10% | 30% | 100% | Average |
| ConCL | 0.6773 | 0.7054 | 0.8011 | 0.8483 | 0.7580 |
| CA²CL | 0.7032 | **0.7567** | 0.8107 | **0.8548** | 0.7813 |
| DiCoH | **0.7224** | 0.7524 | **0.8124** | 0.8535 | **0.7852** |

methods achieving +2.00% mIoU gains on scarcest setting while consistently yielding the highest (lowest value) average rank; refer to Fig. 2b for a visualization. Furthermore, unlike prior works, Di-CoH consistently outperforms the ImageNet baseline (grey rows) regarding *both* average mIoU and rank. Local consistency methods yield the next best results (i.e., DenseCL (Wang et al., 2021) and CP2 (Wang et al., 2022a)), highlighting the strength of pixel-level objectives for dense prediction tasks. However, the PixPro (Xie et al., 2021) local consistency method ranks significantly worse than the rest. This is likely because PixPro (Xie et al., 2021) constrains its positive pairs to a neighbourhood of a pre-defined size. Furthermore, BYOL (Grill et al., 2020) emerges as the stronger global consistency approach over Moco-v2 (He et al., 2020) for both the 5% and 10% settings. BYOL (Grill et al., 2020), however, does not contrast views, avoiding semantic collisions.

**Comparison to histopathology-specific SSL baselines.** Table 3 compares DiCoH against histopathology-specific baselines, ConCL and CA²CL, using identical fine-tuning settings. ConCL (Yang et al., 2022) uses concept masks for dense prediction learning, while CA²CL (Li et al., 2025) uses a cluster-aware adversarial learning to generate challenging positive pairs. DiCoH achieves the highest averaged mIoU across all finetuning data amounts, with noticeable improvements in the most data-scarce (5%) setting +1.9% over CA²CL and +4.5% over ConCL. Furthermore, Di-

Table 4: Ablation study of DiCoH's components. Refer to Sec. 4.2 for analyses.

(a) Common siamese architecture additions.

| Additions | Clinic | Colon | ETIS | Kvasir | GlaS |
|---|---|---|---|---|---|
| Glob. projector | 0.7104 | 0.6541 | 0.4632 | 0.7928 | 0.7382 |
| Asym. predictor | 0.7495 | 0.6684 | 0.4411 | 0.7882 | 0.7376 |
| None (DiCoH) | **0.7531** | **0.6761** | **0.4794** | **0.7945** | **0.7524** |

(b) Positive pixel-to-pixel maps

| $\mathcal{L}_{similar}$ | $\mathcal{L}_{spatial}$ | Clinic | Colon | ETIS | Kvasir | Average |
|---|---|---|---|---|---|---|
| - | - | 0.6860 | 0.6756 | 0.3484 | 0.7824 | 0.6231 |
| ✓ | - | 0.7281 | 0.6300 | 0.3905 | 0.7839 | 0.6331 |
| ✓ | ✓ | **0.7531** | **0.6761** | **0.4794** | **0.7945** | **0.6758** |

(c) Using pixel-to-image negative pairs.

| Negative Type | Clinic | Colon | ETIS | Kvasir | Average |
|---|---|---|---|---|---|
| Pixel-Pixel | 0.7491 | 0.6512 | 0.4571 | 0.7878 | 0.6613 |
| Pixel-Image (DiCoH) | **0.7531** | **0.6761** | **0.4794** | **0.7945** | **0.6758** |

(d) Hard negative sampling thresholds

| Percentile | Clinic | Colon | ETIS | Kvasir | Average |
|---|---|---|---|---|---|
| $0^{th}$ | **0.7587** | 0.6575 | 0.4408 | 0.7751 | 0.6580 |
| $50^{th}$ | 0.7396 | **0.6881** | 0.3923 | 0.7832 | 0.6508 |
| $75^{th}$ (DiCoH) | 0.7531 | 0.6761 | **0.4794** | **0.7945** | **0.6758** |

CoH consistently outperforms both methods on polyp segmentation benchmarks (Table 1, Table 2), demonstrating robust generalizability.

## 4.2 ABLATION STUDY

**Use Symmetric Architectures.** Table 4a shows the negative impact of introducing asymmetry into DiCoH's online network (labeled as "Asym. predictor"). The conventional asymmetric predictor reduced downstream performance across all datasets. These results align with our analysis in Sec. 3.4 and Fig. 4 validating the design decision to challenge conventional asymmetrical architectures.

**Use Pixel-Image Negatives.** Table 4c demonstrates that using pixel-image instead of the conventional pixel-pixel negative pairs in $\mathcal{L}_{local}$ (Sec. 3.3) helped mitigate semantic collisions, improving downstream segmentation performance by $+1.5\%$. Furthermore, DiCoH uses the same projection network for local (pixel-level) and global (image-level) representations to stabilize the pixel-image contrastive objective. Table 4a also demonstrates that using a separate network ("Glob. projector"(Wang et al., 2021; Pang et al., 2024)) is detrimental. For instance, with the Clinic dataset, performance drops by more than 4% in mIoU.

**Maintain Focus on Hard Negatives.** Table 4d communicates that DiCoH use of the *hard* (i.e., upper quartile) of pixel-image negative samples improved overall performance by $\sim 2\%$. Avoiding direct contrast between pixel-pixel negatives and instead using pixel-image pairs allows DiCoH to use hard negative sampling since there will be less false negative pairs which would often register as hard negatives (refer to Sec. 3.3).

**Diversify Positive Pixel-Pixel Alignments.** Table 4b shows the compounding effect of combining spatial similarity-based alignment (Sec. 3.2). We observe a significant increase in segmentation strength, $+4\%$, communicating the improved effectiveness of the pretraining.

## 5 CONCLUSION

In this work, we introduced DiCoH, a self-supervised pretraining framework tailored for homogeneous medical datasets. By diversifying pixel-to-pixel positive alignments, replacing pixel-to-pixel negatives with conservative pixel-to-image sampling, and adopting a symmetric siamese architecture, DiCoH directly addresses the pitfalls of existing SSL methods. Across five segmentation benchmarks under severe label scarcity (5–10% labels), DiCoH consistently ranked first, achieving up to +2.0% mIoU improvements over both general-purpose and domain-specific baselines. Our findings demonstrate that careful adaptation of SSL to homogeneous domains is critical for maximizing segmentation performance under realistic annotation constraints. While we focused on CNN backbones and medical imaging, extending DiCoH to non-medical homogeneous domains (e.g., manufacturing, agriculture) and to Transformer architectures remains future work. By clarifying the design choices and benchmarking SSL methods under extreme low-label settings, we hope DiCoH provides a strong baseline and a foundation for more data-efficient pretraining in domains where labeled data is scarce.

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

# A APPENDIX

To investigate the underlying mechanisms of our proposed components, we analyzed the frozen feature spaces of DiCoH and CP2 on the Kvasir training dataset (Figure 5a, 5b) and (Table 6). We also present further analysis in Table 5 (Addressing Asymmetry) and Table 7 (Addressing Pixel-to-image negatives).

A. Mechanism of Symmetry: Preventing Variance Collapse

In homogeneous domains, asymmetric predictors fail to maintain feature diversity. Table 5 compares feature statistics (Feature variance, positive and negative similarity scores) at the end of pretraining to downstream performance. In both Histopathology and Gastrointestinal domains, the asymmetric baseline suffers from **Variance Collapse** (low $\sigma^2$). This collapse forces the model to pull "negative" pairs closer together, resulting in a high Negative Similarity score of 0.291 (Gastrointestinal). This suggests the model is confusing distinct regions. DiCoH improves the variance ($\sigma^2 = 0.087$) and pushes negative similarity down to -0.0016. This pattern is similar across both medical domains and reflects in mIoU improvements from $0.7376 \rightarrow 0.7524$ (Histopathology) and $0.6620 \rightarrow 0.6760$ (Gastrointestinal)

Table 5: **Impact of Symmetry.** We compare feature statistics at the end of pretraining against downstream segmentation (mIoU). In both Histopathology and Gastrointestinal domains, the asymmetric baseline suffers from **Feature Variance Collapse** (low $\sigma^2$) and high **Negative Similarity** (confusion). Restoring symmetry (DiCoH) recovers variance and separates negative pairs, directly correlating with improved mIoU.

| Method | Feat. Var ($\sigma^2$) | Pos. Sim ($\uparrow$) | Neg. Sim ($\downarrow$) | mIoU |
|---|---|---|---|---|
| *Histopathology (GlaS)* | | | | |
| Asymmetric Predictor | 0.039 | 0.554 | 0.202 | 0.7376 |
| **DiCoH (Symmetric)** | **0.087** | **0.701** | **−0.00154** | **0.7524** |
| *Gastrointestinal (Avg)* | | | | |
| Asymmetric Predictor | 0.029 | 0.631 | 0.291 | 0.6620 |
| **DiCoH (Symmetric)** | **0.087** | **0.851** | **−0.00157** | **0.6760** |

To verify that the variance preserved by DiCoH is discriminative (addressing the question of representation geometry), we evaluated the linear separability of pixel embeddings (Polyp vs. Background) using a Linear SVM and Silhouette Score (Table 6). We observe that the asymmetric baseline achieves only 73.0%linear separation accuracy, confirming that the features are entangled. DiCoH improves this to 76.6%(+3.6 points), validating that our symmetric strategy aids to disentangle the semantic classes, making them linearly separable for the downstream segmentation head.

Table 6: **Quantifying Semantic Separation.** We evaluate the linear separability of pixel embeddings (Polyp vs. Background) on Kvasir using a Linear SVM and Silhouette Score. The asymmetric baseline shows lower separability, confirming semantic collision. DiCoH (Symmetric) improves linear accuracy by **+3.6%** and cluster distinctness (Silhouette) by **+0.048**, proving it learns more discriminative boundaries.

| Method | Silhouette Score ($\uparrow$) | Linear Separability (Acc. $\uparrow$) |
|---|---|---|
| Asymmetric Predictor | 0.196 | 73.0% |
| **DiCoH (Symmetric)** | **0.244** | **76.6%** |

B. MECHANISM OF NEGATIVES: PREVENTING SEMANTIC COLLISION

Table 7 compares negative sampling strategies at the end of pre-training. We observe that pixel-to-pixel yields a positive, negative similarity score (0.0023). This indicates semantic collision, that is, the model is being forced to push apart pixels that are actually semantically similar (False negatives). Our pixel-to-image strategy lowers this score to -0.00157 successfully separating negative pairs and reducing false collisions.

Table 7: **Impact of Negative Sampling Strategy.** Pixel-to-Pixel negatives result in positive mean similarity (0.0023) between negative pairs, indicative of *semantic collision*. Our Pixel-to-Image strategy reduces this to $-0.0016$, confirming improved semantic separation.

| Method | Pos. Sim ($\uparrow$) | Neg. Sim ($\downarrow$) | mIoU |
|---|---|---|---|
| Pixel-Pixel | 0.780 | 0.00233 | 0.6613 |
| **Pixel-Image (DiCoH)** | **0.851** | $-0.00157$ | **0.6758** |

C. MECHANISM OF POSITIVES: PREVENTING TRIVIALITY

Standard spatial alignment allows models to minimize loss via trivial local matching. (Figure 5a, 5b) visualizes the self-similarity when querying a pixel on a random polyp image in Kvasir-SEG for pre-trained CP2 and DiCoH. For CP2, the similarity map is localized and fails to activate the upper lobe of the polyp. This indicates the model has not learned the global concept of the object. For DiCoH, the Similarity map activates the entire polyp structure. The queried bottom bulb is linked to the upper lobe. This shows that Diverse Positive Alignment in DiCoH forces the model to learn Semantic Consistency and learn overall object representations rather than local texture patches.

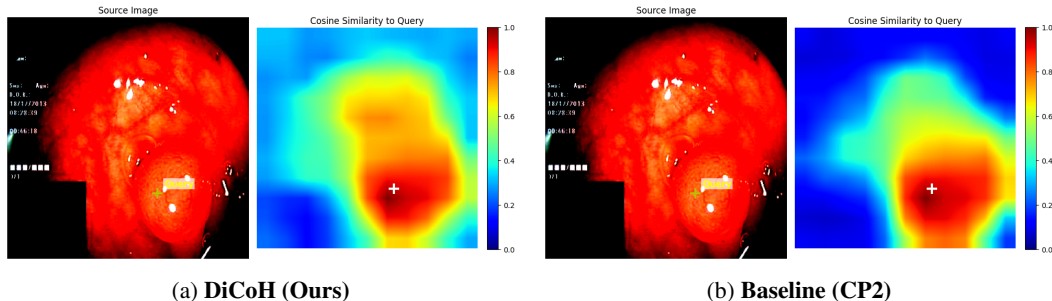

(a) **DiCoH (Ours)**           (b) **Baseline (CP2)**

Figure 5: **Visualizing alignment of related semantics.** We query a single pixel (white cross) within a polyp to visualize feature similarity.

D. ADDITIONAL ANALYSIS

Table 8: **Sensitivity of $\lambda$ between global and local losses.** DiCoH's performance is an improvement over ImageNet for $\lambda$ values that allow both the global and dense losses to contribute (i.e. $0 < \lambda < 1$). This table reports the finetuning (5% data) mIoU across all 5 datasets and 3 seeds.

| $\lambda$ | 0 | 0.25 | 0.5 | 0.75 | 1 |
|---|---|---|---|---|---|
| Average | 0.5869 | 0.5918 | **0.6303** | 0.5825 | 0.5627 |

Table 9: **Performance impact of the negative queue size** $\mathcal{K}$**.** We adopted a queue size of 65536, just like CP2, for fairness and to limit the GPU load. We observe a positive correlation between size and performance which aligns with common observations in contrastive learning (more negative diversity helps). This table reports the finetuning (5% data) mIoU across all 5 datasets and 3 seeds.

| Negative Queue Size | 500 | 5000 | 65536 |
|---|---|---|---|
| Average | 0.5982 | 0.6124 | **0.6303** |

Table 10: **ImageNet-initialization + SSL improves upon random initialization + SSL.** ImageNet-initialization + SSL is a better and practical contribution to medical SSL research. ImageNet-initialization provides a better starting point for pretraining than Random, achieving significantly higher finetuning mIoUs regardless of pretraining strategy, even with 10 times more training. By initializing all methods with ImageNet weights, we ensure fair comparison. If we trained from random, methods that converge slowly (e.g., MoCo) would be unfairly penalized. This table reports the finetuning (5% data) mIoU across all 5 datasets and 3 seeds.

| Pretraining Initialization | BYOL | CP2 | DenseCL | DiCoH | MoCo-v2 |
|---|---|---|---|---|---|
| ImageNet-Initialized | 0.5951 | 0.6075 | 0.6103 | **0.6303** | 0.5853 |
| Random | 0.2453 | 0.2726 | **0.2846** | 0.2548 | 0.2583 |
| Random (x10 Epochs) | 0.3170 | **0.4343** | 0.3104 | 0.3311 | 0.3864 |
| Random (x100 Epochs) | 0.4725 | 0.4847 | 0.4634 | **0.4914** | 0.4843 |

Table 11: **Downstream segmentation performance of publicly released ImageNet-pretrained models.** This table reports the finetuning (100% data) mIoU across all 5 datasets and 3 seeds.

| ImageNet Pretraining | DINOv2 | VICRegL | Barlow | MoCo-v2 | CP2 |
|---|---|---|---|---|---|
| Gastrointestinal (Avg) | 0.8098 | 0.7645 | 0.7713 | 0.8341 | 0.8463 |
| Histopathology (GlaS) | 0.8244 | 0.7412 | 0.7482 | 0.8590 | 0.8595 |

Table 12: **Comparison with Masked-Image-Modelling (MIM) methods.** Finetuning mIoU (5% and 10% data) across five datasets and three seeds for SimMIM (Xie et al., 2022) and SparK (Tian et al., 2023). DiCoH consistently outperforms MIM-based pretraining since MIM benefits most from object-centric natural images, where global scene context helps the model infer the content of masked regions. In homogeneous medical images, however, the absence of strong object boundaries and the presence of fine-grained, repetitive tissue textures make masked-region prediction either trivial (easy to guess) or poorly conditioned (multiple plausible completions). As a result, MIM yields weaker and less discriminative learning signals compared to DiCoH. Furthermore prior work (e.g., Woo et al. (2023)) notes that MIM is less effective on CNN backbones, limiting its usefulness in medical imaging where CNNs remain popular. Note all methods use ResNet-50 backbones for fair comparison.

| Data Size | Method | Gastrointestinal | | | | Histopathology | Average |
|---|---|---|---|---|---|---|---|
| | | Clinic | Colon | Etis | Kvasir | GlaS | |
| | SimMIM | 0.6394 | 0.5300 | 0.3652 | 0.7282 | 0.6524 | 0.5830 |
| 5% | SparK | 0.6514 | 0.4823 | 0.3565 | 0.6783 | 0.6188 | 0.5575 |
| | DiCoH | **0.6702** | **0.5603** | **0.4526** | **0.7458** | **0.7224** | **0.6303** |
| | SimMIM | 0.7288 | 0.6240 | 0.5156 | 0.7669 | 0.7150 | 0.6701 |
| 10% | SparK | 0.7128 | 0.5820 | 0.4458 | 0.7572 | 0.7188 | 0.6433 |
| | DiCoH | **0.7531** | **0.6761** | **0.4794** | **0.7945** | **0.7524** | **0.6911** |

Table 13: **Evaluating SSL pertaining methods on gastrointestinal (polyp segmentation) and histopathology (gland segmentation) datasets using 5% of the finetuning data.** Note that **bold** numbers denotes best; underline, second. We report standard deviation values.

| Method | Gastrointestinal | | | | Histopathology | Average |
|---|---|---|---|---|---|---|
| | Clinic | Colon | Etis | Kvasir | GlaS | |
| ImageNet | 0.6676±0.01 | 0.5400±0.06 | 0.3043±0.03 | 0.7392±0.02 | 0.6619±0.05 | 0.5826 |
| BYOL | 0.6902±0.02 | 0.5373±0.03 | 0.3009±0.07 | 0.7469±0.02 | 0.7003±0.02 | 0.5951 |
| MoCo-v2 | 0.6831±0.02 | 0.4836±0.08 | 0.3207±0.05 | 0.7419±0.03 | 0.6973±0.01 | 0.5853 |
| DenseCL | 0.6673±0.02 | 0.6331±0.02 | 0.3238±0.14 | 0.7320±0.01 | 0.6953±0.01 | 0.6103 |
| PixPro | 0.6316±0.02 | 0.5974±0.03 | 0.3706±0.07 | 0.6978±0.01 | 0.6497±0.04 | 0.5894 |
| CP2 | 0.6524±0.01 | 0.6066±0.05 | 0.3918±0.03 | 0.7491±0.02 | 0.6374±0.07 | 0.6075 |
| ConCL | 0.6752±0.01 | 0.5205±0.05 | 0.3304±0.19 | 0.7231±0.04 | 0.6773±0.01 | 0.5853 |
| CA2CL | 0.6654±0.01 | 0.5135±0.04 | 0.3893±0.10 | 0.7215±0.01 | 0.7032±0.03 | 0.5986 |
| DiCoH | 0.6702±0.03 | 0.5603±0.08 | 0.4526±0.11 | 0.7458±0.01 | 0.7224±0.01 | **0.6303** |

