# OpenReview forum: "DiCoH: Rethinking Self-Supervised Pretraining for Semantic Segmentation in Homogenous Medical Domains"
_ICLR.cc/2026/Conference — Submitted to ICLR 2026_

### Official Review · Reviewer_jHrq · 2025-10-18

**Soundness:** 2
**Presentation:** 3
**Contribution:** 3
**Rating:** 4
**Confidence:** 3

**Summary:**

This paper proposes DiCoH (Diverse Contrastive Learning for Homogeneous Data), a self-supervised pretraining method for semantic segmentation in homogeneous medical images. Existing contrastive learning methods such as SimCLR or MoCo work well for natural images but can perform poorly on medical data where images are visually similar and lack clear boundaries. The paper identifies three main problems in such data: trivial positive pairs, false negatives caused by semantic collisions, and feature variance collapse from asymmetric architectures.

To address these, the proposed method DiCoH introduces (1) diverse positive alignment combining spatial and similarity-based pixel matches, (2) pixel-to-image negative sampling to avoid semantic overlap, and (3) a symmetric siamese architecture that preserves feature variance. The model is trained with a novel joint objective which is specially designed to balance local and global contrastive losses.
On five medical segmentation datasets, DiCoH outperforms prior self-supervised and supervised baselines, achieving up to about two percentage points higher mean IoU under low-data conditions (5–10% labeled samples). The results show that adapting SSL designs to homogeneous domains can significantly improve label-efficient medical image segmentation.

**Strengths:**

**Originality**:

The paper considers an important problem—SSL on homogeneous datasets—and introduces a new strategy combining diverse pixel alignments, pixel-to-image negatives, and symmetry. The idea of using pixel-to-image negatives seems novel.

**Quality**:
The methodology is rigorous and well-presented, including ablations isolating each of the 3 key components (alignment, negatives, architecture). Experimental design which uses ImageNet-initialized backbones and consistent evaluation across five datasets seems fair.

**Clarity**:
The paper is well-organized with clear figures. Mathematical notation is clear. It is especially good that the motivation for each key design is directly supported by empirical findings (e.g., Fig. 4’s variance collapse evidence).

**Significance**:
The contribution is meaningful for medical imaging and potentially for other homogeneous domains (e.g., agriculture, manufacturing). Improving label efficiency in such areas can have high real-world impact. The work also challenges an established design norm—architectural asymmetry—in contrastive learning.

**Weaknesses:**

(1) The paper presents a clear motivation and shows consistent improvements. But I feel a major weakness is that its empirical analysis lacks sufficient depth. The experiments demonstrate that DiCoH performs better than prior methods, but they do not fully explain why each of the three components (diverse positive alignment, pixel-to-image negatives, and symmetricity) contributes to the gain or how these components interact. For example, the analysis mostly reports performance differences gained from the three components without probing the underlying behavior of representations or learning dynamics.

(2) Architecture Choice:
The use of a ResNet-50 + DeepLabv3 backbone is fine but somewhat quite old. Evaluating DiCoH on newer architecture such as Vision Transformers might be a better choice? It can also demonstrate the adaptivity of DiCoH to various architectures.

(3) While the authors mention potential generalization beyond medical data, experiments are restricted to medical segmentation. Validation on non-medical homogeneous datasets (e.g., industrial defect detection) would strengthen claims of general applicability.

(4) The reported performance improvement seems to lack a statistical confidence interval (i.e. error bar).

**Questions:**

(1) How sensitive is DiCoH’s performance to the weighting parameter $\lambda$ between global and local losses?

(2) How large is the negative image queue, and how does its size affect the performance of the method?

(3) The paper claims that symmetric architectures “preserve feature variance.” But it seems unclear how this translates to better representation geometry? In there any quantitative metric that can demonstrate that?

(4) What is  the computational cost of pixel-to-image negatives and dual alignment strategies? Is it huge?

---

> ### Author Response · Authors · 2025-11-22
>
> We thank the reviewer for providing detailed feedback which made engaging with your reviews significantly easier. We categorized them, and addressed them below.
>
> ---
>
> ***Addressing [W1]  … clear motivation… consistent improvements… do not fully explain why each of the three components (diverse positive alignment, pixel-to-image negatives, and symmetricity) contributes to the gain.. [Q3] ..any quantitative metric that can demonstrate symmetric architectures “preserve feature variance.”***
>
> We agree that while our ablation studies demonstrated performance gains, it would indeed be helpful to further visualize why standard methods fail in homogenous settings. To address this we have included furthere analyses in the Appendix (Tables 5, 6, 7 and Figure 5). We present a brief summary of each component here:
>
> 1.  **Addressing Symmetry (Preventing Variance Collapse) [Appendix Table 5 and 6]:**
>     - Contrasting with findings in natural domains (ImageNet) where asymmetry is useful to preserve variance [1], we observe that this relationship is flipped in homogenous settings. Our findings how that the asymmetric baseline suffers from feature variance collapse ($\sigma^2$ = 0.877 to 0.029 for Gastrointestinal) forces the model to confuse distinct regions, increasing negative pixel-to-pixel similarity scores (-0.0016 to 0.291). To better answer **[Q3]** we also train a linear SVM on frozen features to demonstrate the impact.
> 2. **Pixel-to-Image Negatives (Preventing Collision) [Appendix Table 7]**
>     - We compare the standard Pixel-to-Pixel negatives against our Pixel-to-Image (DiCoH) strategy. The standard approach yields a positive mean similarity (0.0023) between negative pairs indicating semantic collisions. The pixel-to-image strategy lowers this score to -0.00157 successfully separating negative pairs and reducing false collisions.
> 3. **Diverse-Positives [Appendix Figure 5]**
>     - We visualize the self-similarity heatmaps by querying a single polyp pixel. Baselines like CP2, produce a highly localized heatmaps and fails to activate the upper lobe of the polyp. In contrast, DiCoH produces a heatmap that activates the entire polyp structure, linking the queried bulb to the upper lobe. This demonstrates DiCoH's strong ability to recognize related concepts.
>
> ---
>
> ***Addressing [Q1] Sensitivity of $\lambda$ between global and local losses?***
>
> DiCoH’s performance is an improvement over ImageNet for $\lambda$ values that allow both the global and dense losses to contribute (i.e. $0<\lambda<1$). This table is now in the Appendix.
>
> *[Table] Finetune (5% data) mIoU across all 5 datasets and 3 seeds*
>
> | 0 | 0.25 | 0.5 | 0.75 | 1 | ImageNet |
> | --- | --- | --- | --- | --- | --- |
> | 0.5869 | 0.5918 | **0.6303** | 0.5825 | 0.5627 | 0.5826 |
>
> ---
>
> ***Addressing [Q2] How large is the negative image queue … affect the performance***
>
> We adopted a queue size of 65536, just like CP2, for fairness and to limit the GPU load. We observe a positive correlation between size and performance which aligns with common observations in contrastive learning (more negative diversity helps)[4]. We will report these numbers in the appendix.
>
> *[Table] Finetune (5% data) mIoU across all 5 datasets and 3 seeds*
> | Queue Size | 500 | 5000 | 65536 |
> | --- | --- | --- | --- |
> | Average mIoU | 0.5982 | 0.6124 | **0.6303** |
>
> ---
>
> ***Addressing [Q4] ... computational cost of pixel-to-image negatives and dual alignment strategies? Is it huge?***
>
> The pixel-to-image negative strategy is actually half the memory cost of the pixel-to-pixel, because only one image-level negative queue is needed for both the local and global losses. Furthermore, DiCoH is the lightest out of the current dual (pixel+global) alignment strategies
>
> | Method | DiCoH | MIM | CP2 | DenseCL |
> | --- | --- | --- | --- | --- |
> | Parameters | **28M** | 42M | 66M | 33M |
>
> ---
>
> ***Addressing [W2]  …ResNet-50 + DeepLabv3 backbone is fine but … Vision Transformers might be a better choice? … can also demonstrate the adaptivity of DiCoH***
>
> We chose a CNN-based backbone because it is a stable and efficient architecture for low data medical segmentation [5][6], and it allowed direct comparison with prior works that use the same setup. CNN backbones like ResNet still offer strong performance in low-data regimes: they capture useful local features, impose fewer restrictions on input size, and are lightweight to train, attributing to their large-scale use [3]. We agree that Vision Transformers (ViTs) have shown excellent performance in segmentation, but ViTs also tend to require larger training datasets or extensive augmentation due to their lack of built-in inductive bias [3][6]. ViTs also come with higher computational cost. That said, our proposed techniques are architecture-agnostic in principle and we expect DiCoH could likewise improve a ViT-based model, and exploring heterogeneous architectures is an interesting avenue for future work.

---

> > ### Author Response · Authors · 2025-11-22
> >
> > ***Addressing [W4] …lack a statistical confidence interval (i.e. error bar).***
> >
> > We followed the standard protocol of reporting mean performance over 3 independent runs (with different random seeds). With only three samples, formal confidence intervals would not be statistically meaningful (the sample is too small to reliably estimate variance for significance testing). However, to provide the reader with a sense of variability, we will include the standard deviation of mIoU across runs for each result in the Appendix Table 13.
> >
> > ---
> >
> > ***Addressing [W3] While the authors mention potential generalization beyond medical data, experiments are restricted to medical segmentation. Validation on non-medical homogeneous datasets (e.g., industrial defect detection) would strengthen claims of general applicability.***
> >
> > Absolutely! Although the paper particularly focuses on medical datasets, we are committed to releasing code and pretrained weights to enable community evaluation on additional domains.
> >
> > ---
> >
> > [1] Xiao Wang, Haoqi Fan, Yuandong Tian, Daisuke Kihara, and Xinlei Chen. On the importance of asymmetry for siamese representation learning. In Proceedings of the IEEE/CVF CVPR, pp. 16570–16579, 2022b.
> >
> > [2] Yonglong Tian, Chen Sun, Ben Poole, Dilip Krishnan, Cordelia Schmid, and Phillip Isola. *What Makes for Good Views for Contrastive Learning?* In Advances in Neural Information Processing Systems (NeurIPS), 2020
> >
> > [3] Takahashi, Satoshi et al. “Comparison of Vision Transformers and Convolutional Neural Networks in Medical Image Analysis: A Systematic Review.” *Journal of medical systems* vol. 48,1 84. 12 Sep. 2024, doi:10.1007/s10916-024-02105-8
> >
> > [4] He, Kaiming, et al. "Momentum contrast for unsupervised visual representation learning." *Proceedings of the IEEE/CVF conference on computer vision and pattern recognition*. 2020.
> >
> > [5] Li, Jun et al. “Transforming medical imaging with Transformers? A comparative review of key properties, current progresses, and future perspectives.” *Medical image analysis* vol. 85 (2023): 102762. doi:10.1016/j.media.2023.102762
> >
> > [6] Ha, Jungmin, et al. "Leveraging Inductive Bias in ViT for Medical Image Diagnosis." *35th British Machine Vision Conference*, (2024).

---

> ### Comment · Reviewer_jHrq · 2025-11-26
> **Re: rebuttal**
>
> I thank the authors for their effort in preparing the rebuttal.
> Their rebuttal addressed my questions regarding: (1) sensitivity of performance w.r.t. $\lambda$; (2) size of image queue; (3) computational cost of pixel-to-image negatives and dual alignment strategies.
>
> Regarding the remaining weaknesses: (1) it is understandable (though not ideal) that a statistical error bar is missing with 3 independent trials; (2) further analysis has been added to disentangle the performance contribution of the three proposed components.
>
> Given the response, I raise my score to 6 for now. I will also take into consideration other reviewers's comments and corresponding rebuttals during the discussion stage.

---

> > ### Author Response · Authors · 2025-11-26
> >
> > We sincerely thank the reviewer for their effort reviewing our paper and providing us an opportunity to address their concerns,  and in turn strengthening our papers clarity!

---

### Official Review · Reviewer_yTef · 2025-10-30

**Soundness:** 2
**Presentation:** 3
**Contribution:** 2
**Rating:** 4
**Confidence:** 4

**Summary:**

The authors propose a self-supervised learning approach, DICOH for pre-training deep learning models for dense tasks in medical imaging contexts, where data is more homogeneous than in computer vision. DICOH is a contrastive approach with global (instance-level) and local (pixel-level) losses that aims to avoid trivial positive pairs (by defining a positive based on spatial correspondence and another one based on feature cosine similarity), false negative pairs (by adopting a pixel-to-image negative strategy), and that uses a symmetric siamese architecture. The proposed method is compared against various instance-level pre-training strategies, semi-local contrastive strategies, and histopathology SSL methods (ConCL, CA2CL), assuming ImageNet-based pre-retraining, showing generally good performance.

**Strengths:**

1. The proposed approach shows promising performance in low data settings.
2. The ablation study shows the impact of specific elements introduced by the proposed approach
3. Clarity: The paper is easy to read and generally clear.

**Weaknesses:**

1. I have reservations regarding some of the main contributions:

Contribution 1. Actually no "analysis" is provided as to "why trivial positives, false negatives, and asymmetric designs degrade contrastive pretraining in homogeneous medical data", only an ablation study that supports this statement.

Contribution 2. "Diversified positives (spatial + similarity)" were already introduced in VICRegL, Bardes et al. 2022. The proposed method is not compared against VICRegL. "Pixel-to-image negatives" were already proposed in DenseCL. The "symmetric siamese design" is the standard in contrastive approaches, to which the proposed method belong, rather than a novelty. (Asymmetric designs are more commonly used in self-distillation or joint embeddings SSL)

Contribution 3. I have reservations here. Indeed, dense SSL benchmarks in the medical domain in annotation scarce regimes are generally not conducted under an ImageNet-pre-pretraining. But it is not clear 1) whether this ImageNet-initialization + SSL improves upon random initialization + SSL across most medical domains, due to the domain gap. 2) whether ImageNet-Initialization should be seen as a positive (fairer) or negative (detrimental) change to the baselines the authors compare against. It would be helpful to see the performance of each method, including the proposed method, without this initialization. This is also a limitation to the scope of the paper, as this type of initialization is very specific to 2D applications, whereas many medical domains are 3D.

2. A "pure" pixel-level SSL baseline defining positive pairs from the natural correspondence maps after augmentations, coming from the medical domain, for example vox2vec [1], is missing.

3. A dense representation baseline coming from foundation models, such as DINOv2 or DINOv3, would also be a welcome addition.

4. The impact of ImageNet-initialization is not ablated.

5. It is not clear whether baselines are trained using their optimal hyperparameter settings or the same settings as for the proposed method (regarding learning rates, batch size, choice and strengths of augmentations, etc.), which have probably been fine-tuned specifically for the proposed method. Could the authors kindly clarify ?


[1] vox2vec: A Framework for Self-supervised Contrastive Learning of Voxel-level Representations in Medical Images, Goncharov et al. MICCAI 2023

**Questions:**

The comments and questions raised in the weaknesses section above are the main factors in my initial rating.

---

> ### Author Response · Authors · 2025-11-22
>
> We thank the reviewer for providing detailed feedback which made engaging with you reviews significantly easier. We categorized them and addressed them below.
>
> ---
>
> ***Addressing [W1.1] Contribution 1. Actually no "analysis" is provided as to "why trivial positives, false negatives, and asymmetric designs degrade contrastive pretraining in homogeneous medical data", only an ablation study that supports this statements***
>
> We thank the reviewer for this critical insight.  To better address our claims we have included the full quantitive and qualitative analysis in Appendix A (Tables 5-7, Figure 5).
>
> 1.  **Analyzing Asymmetric Design [Appendix Table 5 and 6]:**
>     - Contrasting with findings in natural domains (ImageNet) where asymmetry is useful to preserve variance [1], we observe that this relationship is flipped in homogenous settings. Our findings how that the asymmetric baseline suffers from feature variance collapse ($\sigma^2$ = 0.877 to 0.029 for Gastrointestinal) forces the model to confuse distinct regions, increasing negative pixel-to-pixel similarity scores (-0.0016 to 0.291). We also train a linear SVM on frozen features to demonstrate the impact.
> 2. **Analyzing False Negatives & Semantic Collision [Appendix Table 7]**
>     - We compare the standard Pixel-to-Pixel negatives against our Pixel-to-Image (DiCoH) strategy. The standard approach yields a positive mean similarity (0.0023) between negative pairs indicating semantic collisions. The pixel-to-image strategy lowers this score to -0.00157 successfully separating negative pairs and reducing false collisions.
> 3. **Analyzing Trivial Positives [Appendix Figure 5]**
>     - We visualize the self-similarity heatmaps by querying a single polyp pixel. Baselines like CP2, produce a highly localized heatmaps and fails to activate the upper lobe of the polyp. In contrast, DiCoH produces a heatmap that activates the entire polyp structure, linking the queried bulb to the upper lobe. This demonstrates DiCoH's strong ability to recognize related concepts.
>
> ---
>
> ***Addressing [W4][W1.3]  Not clear whether ImageNet-initialization + SSL improves upon random initialization + SSL. Would be helpful to see the performance of each method without***
>
> Thank you for highlighting this important design decision. ImageNet-initialization provides a better starting point for pretraining than Random, achieving significantly higher finetuning mIoUs regardless of pretraining strategy, even with 10 times more pretraining; analysis added to Appendix Table 10.
>
> *[Table] Finetune (5% data) mIoU across all 5 datasets with/without ImageNet-initialization before further SSL pretraining*
> | Pretraining Initialization | BYOL | CP2 | DenseCL | DiCoH | MoCo-v2 |
> | --- | --- | --- | --- | --- | --- |
> | ImageNet-Initialized | **0.5951** | **0.6075** | **0.6103** | **0.6303** | **0.5853** |
> | Random | 0.2453 | 0.2726 | 0.2846 | 0.2548 | 0.2583 |
> | Random (x10 Epochs)  | 0.3170 | 0.4343 | 0.3104 | 0.3311 | 0.3864 |
>
> **Improvements extend to 3D as well:** The benefits of natural-image pretraining are also observed for 3D volumetric imaging observed in (Ke, Alexander, et al. "Video pretraining advances 3D deep learning on chest CT tasks." *Medical Imaging with Deep Learning*. PMLR, 2024)
>
> ---
>
> ***Addressing [W2] A "pure" pixel-level SSL baseline… like vox2vec [1]***
>
> While Vox2Vec is designed for 3D voxel data, adapting 3D SSL methods to 2D is non-trivial due to architectural mismatches [2]; we instead compare against CA2CL (2025), which outperforms earlier pure pixel-level, spatial-correspondence approaches like PLM-SSL [3]. To directly validate our hypothesis that pixel-only correspondence is insufficient, we also now include an ablation (Pixel-Only + Spatial-Only) that removes semantic similarity and image-level negatives.
>
> *[Table] Finetune (5% data) mIoU across all 5 datasets and 3 seeds*
>
> | Pretraining Approach | Clinic | Colon | ETIS | Kvasir | GlaS | Average |
> | --- | --- | --- | --- | --- | --- | --- |
> | PixelOnly-SpatialOnly | 0.6309 | 0.5198 | 0.3344 | 0.6076 | 0.7081 | 0.5601 |
> | DiCoH | **0.6702** | **0.5603** | **0.4526** | **0.7458** | **0.7224** | **0.6303** |
>
> ---
>
> ***Addressing [W3]  … baseline… foundation models, … DINOv2 or DINOv3, …  be a welcome addition. [W1.2] VICRegL***
>
> Indeed! We evaluated the released ImageNet-pretrained weights of DINOv2, VICRegL, and Barlow Twins. As Appendix Table 11 shows, they transfer considerably worse than MoCo-v2 and CP2 on medical segmentation. For this reason, we focused on stronger, more relevant baselines when evaluating DiCoH.
>
> Evaluating the released ImageNet data pretrained weights of different SSL methods on 100% finetuning data  across all 5 datasets and 3 seeds
>
> | ImageNet Data Pretraining | DINOv2 | VICRegL | Barlow | MoCo-v2 | CP2 |
> | --- | --- | --- | --- | --- | --- |
> | Polyp. | 0.8098 | 0.7645 | 0.7713 | 0.8341 | 0.8463 |
> | Hist. | 0.8244 | 0.7412 | 0.7482 | 0.8590 | 0.8595 |

---

> > ### Author Response · Authors · 2025-11-22
> >
> > ***Addressing [W1.2] Novelty concerns***
> >
> > Our contribution is (1) identifying *that/why* standard SSL practices fail on homogeneous
> > data (Section 1, challenges i-iii) and (2) proposing a unified framework that
> > addresses all three simultaneously; we've expanded our quantitive and qualitative analysis in the Appendix.
> >
> > 1. **Regarding symmetry**: Prior work has not shown that asymmetric architectures (beneficial in natural-image SSL [1]) consistently harm representation quality in homogeneous medical domains.
> >
> > 2. **Regarding pixel-to-image:**  DenseCL only employs image-level negatives because it’s (direct quote) “conceptually simpler”, while this paper demonstrates why it is actually significant for learning meaningful representations for homogenous medical data (Table 4c).
> >
> > 3. **Regarding VICRegL**: Although it uses diversified positives, its released ImageNet-pretrained weights performed substantially worse than CP2, MoCo-v2, and even Barlow Twins (another non-contrastive, covariance-based method) on our medical segmentation tasks (Appendix Table 11). Given its weak starting point and high retraining cost, we prioritized CP2, a strong pixel-level baseline that also uses spatial/similarity positives.
> >
> > ---
> >
> > ***Addressing [W5] Are baselines trained with their optimal hyperparameter settings or the same as DiCoH…***
> >
> > Happy to clarify! **We did not optimize the hyperparameters for DiCoH.** Apart from halving the batch size (256-128) to fit our GPUs (meaning we needed to drop the learning rate to 0.001), we used common hyper-parameters which are quite consistent across SSL methods. Namely 0.0001 weight decay, SGD optimizer, and basic jitter, blurring, cropping and flipping augmentations (CP2,MoCo,…). However, we faithfully allowed specialized methods like CONCL and CA2CL to deploy their own custom augmentations since that is part of their contribution. Fine-tuning was consistent across methods for fair evaluation. We even matched our negative queue size to the CP2 paper for fair evaluation.
> >
> > We will also be releasing our code for reproducibility.
> >
> > ---
> >
> > [1] Xiao Wang, Haoqi Fan, Yuandong Tian, Daisuke Kihara, and Xinlei Chen. On the importance of asymmetry for siamese representation learning. In Proceedings of the IEEE/CVF CVPR, pp. 16570–16579, 2022b.
> >
> > [2] Ali, Mudassar, et al. "A review of the segment anything model (sam) for medical image analysis: Accomplishments and perspectives." *Computerized Medical Imaging and Graphics*119 (2025): 102473.
> >
> > [3] Zhang, Jingwei, et al. "Precise location matching improves dense contrastive learning in digital pathology." *International Conference on Information Processing in Medical Imaging*. Cham: Springer Nature Switzerland, 2023.

---

> > > ### Comment · Reviewer_yTef · 2025-11-26
> > >
> > > Many thanks to the authors for their detailed responses to the reviewers' comments. In light of all added information, I still have a few reservations about the results and about the contributions that inform my score.
> > >
> > > _____
> > >
> > > Regarding results, the method's performance seems to heavily depend on the initialization (cf. Table 10). With random initialization instead of ImageNet, DiCoH is behind 2 of the 4 chosen baselines: CP2 and MoCo-v2 (and is only on par with a 3rd, DenseCL). This sensitivity also weakens the rationale behind some of the design choices of DiCoH.
> > >
> > > Moreover, in the main Tables (1 and 2) on many tasks, the method is not significantly above the best competitor. In Table 2 and Fig 2.b, the results are somewhat misleading, as the Average Rank of the baseline CA2CL is 3.8, i.e. worse, despite a better (or equally good) Average mIoU of 0.6920 vs. Average Rank of DiCoH is 1.8 with lower Average mIoU (0.6911).
> > >
> > > _____
> > >
> > > Regarding contributions, in my view the paper brings together ideas that were introduced previously in the literature in the exact same context, and shows (some) performance improvements empirically, specifically on histopathology and gastrointestinal datasets. However, this is not how the paper frames itself.
> > >
> > > Furthermore I am still unconvinced by the claim that the paper "challenge[s] conventional asymmetrical architectures". In my view, a large number of contrastive learning papers use a symmetrical design, including the seminal papers SimCLR and MoCo and other early dense representation papers such as [2], [3] and PixPro, etc. Is this statement strictly based on [1]?
> > >
> > > Finally, I acknowledge that DenseCL authors state that pixel-to-image negatives are "conceptually simpler", whereas in the present paper, the authors propose an ablation study of pixel-to-image vs. pixel-to-pixel negatives. But it is unclear to me whether an ablation study regarding an earlier contribution is a strong enough contribution in itself.
> > > In particular, the paper does not offer a theoretical analysis of "why" pixel-to-image negatives would be superior in general, beyond limited insights in Table 7 into the empirical mean positive/negative similarity (where it is unclear to me whether the comparison highlights statistically significant differences).
> > >
> > > _____
> > >
> > > [1] Xiao Wang, Haoqi Fan, Yuandong Tian, Daisuke Kihara, and Xinlei Chen. On the importance of asymmetry for siamese representation learning. In Proceedings of the IEEE/CVF CVPR, pp. 16570–16579, 2022b.
> > >
> > > [2] Pinheiro, P. O., et al. "Unsupervised learning of dense visual representations." NeuRIPS (2020)
> > >
> > > [3] Chaitanya, K., et al. "Contrastive learning of global and local features for medical image segmentation with limited annotations." NeuRIPS (2020)

---

> ### Author Response · Authors · 2025-11-27
> **Part 1/2**
>
> [Part 1/2] We sincerely thank the reviewer for continuing to engage with our research! We hope to continue addressing the remaining reservations.
>
> ---
>
> **Addressing | *“…the method's performance heavily depends on the initialization…”***
>
> We appreciate this important point about initialization. In our initial rebuttal we only reported short random-initialized runs; this is not enough time for all pretraining methods to converge when trained from scratch. After extending random-initialized pretraining by x100 epochs, DiCoH is in fact ahead. This is why recent medical papers advocate initializing from ImageNet pretrained weights [1][2]; (1) better results and (2) convergence across methods with less epochs.
>
> | Pretraining Initialization | Spread | BYOL | CP2 | DenseCL | DiCoH | MoCo-v2 |
> | --- | --- | --- | --- | --- | --- | --- |
> | ImageNet-Initialized | 0.045 | 0.5951 | 0.6075 | 0.6103 | **0.6303** | 0.5853 |
> | Random | 0.0393 | 0.2453 | 0.2726 | **0.2846** | 0.2548 | 0.2583 |
> | Random (x10 Epochs)  | 0.1239 | 0.3170 | **0.4343** | 0.3104 | 0.3311 | 0.3864 |
> | Random (x100 Epochs) | 0.028 | 0.4725 | 0.4847 | 0.4634 | **0.4914** | 0.4843 |
>
> We’ve now updated Table 10 to show this. We could try squeeze in more training to demonstrate consistent improvements if that would be convincing?
>
> ---
>
> **Addressing | *“…method is not significantly above the best competitor…results are somewhat misleading, as the Average Rank of the baseline CA2CL is 3.8 (worse) despite a better (or equally good) Average mIoU…”***
>
> We thank you for this critical observation. The 0.09% mIoU gap is indeed marginal and therefore reveals precisely **why average rank is more informative for multi-dataset comparison**. In fact, the reviewer will see that this SSL paper is unique in showing this average rank analysis.
>
> *[Table] Finetuning win rate (10% data, Table 2) across all 5 datasets (3 seeds)*
>
> | Dataset | Win Rate | mIoU | Avg. Rank | Dataset Ranks |
> | --- | --- | --- | --- | --- |
> | CA2CL | 20% (1/5) | 0.6920 | 3.8 | 7,5,1,4,2 |
> | DiCoH | 60% (3/5) | 0.6911 | 1.8 | 1,1,3,1,3 |
>
> **The story:** CA2CL's +0.09% mIoU comes from a single dominant ETIS result (0.5347, +4% over DiCoH). Meanwhile, CA2CL ranks 7th on Clinic and 5th on Colon. These weaknesses are completely hidden by the averaged mIoU. In contrast, DiCoH achieves ranks 1–3 on all five datasets and wins 3/5 tasks, indicating more consistent performance across different medical domains.
>
> ***Looking at this table, if deploying to a new, untested medical imaging dataset, which method would you try first?***
>
> This is why the seminal work by **Demšar (2006), "Statistical Comparisons of Classifiers over Multiple Data Sets,"[3] (16531 Citations)** explicitly recommends ranking-based approaches for multi-dataset evaluation.
>
> Our revision will add win rates to Tables 1–2 alongside average mIoU and average rank, so that both peak performance and robustness are explicit. If you feel an additional summary (e.g., a figure of rank distributions) would further clarify this, we are happy to include it!
>
> ---
>
> We hope these satisfy the first part of your reply. We'll respond to the rest shortly
>
> ---
>
> [1] Sanderson, Edward, and Bogdan J. Matuszewski. "A study on self-supervised pretraining for vision problems in gastrointestinal endoscopy." *IEEE Access* 12 (2024): 46181-46201.
>
> [2] VanBerlo, Blake, Jesse Hoey, and Alexander Wong. "A survey of the impact of self-supervised pretraining for diagnostic tasks in medical X-ray, CT, MRI, and ultrasound." *BMC Medical Imaging* 24.1 (2024): 79.
>
> [3] Demšar, Janez. "Statistical comparisons of classifiers over multiple data sets." *Journal of Machine learning research*7.Jan (2006): 1-30.

---

> > ### Author Response · Authors · 2025-11-28
> > **Part 2/2**
> >
> > **Re: "Bringing together existing ideas" and Contribution Framing**
> >
> > We thank the reviewer for this insightful critique. We will refined our claims to clarify that DiCoH is not proposing new architectural blocks, but rather ***providing a validated blueprint for SSL in homogeneous domains***. The current literature lacks consensus: **(1)** some methods use asymmetrical architectures (BYOL/DenseCL…), others don't (MoCo,CP2…); **(2)** some use pixel-to-image negatives, others pixel-to-pixel; **(3)** some use spatial-only (PixPro) positives, others similarity-only (DenseCL,CA2CL), others spatial+similarity (VICRegL), others all-pixels (CP2)….
> >
> > We cut through this ambiguity by showing that in the homogeneous medical data space, symmetry, pixel-to-image negatives, spatial+similarity-based positives are not optional features, but consistently effective. We provide rigorous (multi-dataset/seed/size) empirical guidance to move the field beyond trial-and-error, by linking these design choices directly to failure modes like variance collapse and semantic collisions.
> >
> > Concretely, ***DiCoH does three things that, to our knowledge, prior work has not done in medical domains*** where homogeneity is not the exception but the rule:
> >
> > 1. Identifies and disentangles three concrete failure modes (trivial positives, semantic collisions, and variance collapse) as key obstacles in homogeneous medical SSL.
> > 2. Shows that, in this setting, “mixing and matching” existing ideas is *not* arbitrary: specific design choices are required to reliably combat those failure modes.
> > 3. Provides consolidated, cross‑dataset evidence that is meant to guide future method design in this domain.
> >
> > ---
> >
> > **Re: Asymmetry and "Challenging Conventional Architectures"**
> >
> > We accept the reviewer's correction regarding "challenging conventional asymmetrical architectures" being an overstatement given the prevalence of symmetric designs (e.g., SimCLR, MoCo). We have revised our argument to reflect the precise distinction: ***in natural image SSL, there is no consensus on the necessity of symmetry.*** Because natural data is highly heterogeneous (high variance), methods succeed both with symmetry and with asymmetry (e.g., SimSiam), as the latter’s variance-reducing effects [1] are non-destructive in that domain.
> >
> > However, we demonstrate that this flexibility does not exist in the **homogeneous medical domain**. In this regime, where intrinsic data variance is already low, the variance-reducing properties of asymmetric architectures lead to collapse. Our contribution is establishing that **symmetry is not merely a design alternative, but necessary** to maintain representation variance in homogeneous data. This finding holds true across our exhaustive evaluation of multiple dataset types (histopathology and gastrointestinal), varying label fractions, and multiple random seeds (showing both Average mIoU and Average Rank).
> >
> > ---
> >
> > **Re: Pixel-to-Image Negatives (Theoretical Justification)**
> >
> > The reviewer asks *why* pixel-to-image negatives outperform pixel-to-pixel in this context. Beyond the empirical results in Table (4c & 7), we offer the following analysis in the revision:
> >
> > In natural images, pixel-to-pixel negatives work because objects are distinct (a pixel of a dog is distinct from a pixel of grass). In homogeneous medical data, pixel-to-pixel comparison suffers from **semantic collisions**: two pixels from different images may look identical (e.g., two patches of healthy stroma) but belong to different contexts.
> >
> > - **Pixel-to-Pixel:** Encourages the model to push apart visually similar pixels, effectively teaching it to hallucinate differences in identical textures (false negatives).
> > - **Pixel-to-Image:** Forces the model to distinguish a pixel not from another random pixel, but from a *global context representation*. This anchors the local feature to the global semantic structure, mitigating the homogeneity problem.
> >
> > Taken together, these insights position DiCoH as a practically useful and empirically grounded reference point for developing SSL methods in homogeneous medical domains, moving beyond trial‑and‑error adaptation of SOTA natural‑image methods.
> >
> > ---
> >
> > [1] Xiao Wang, Haoqi Fan, Yuandong Tian, Daisuke Kihara, and Xinlei Chen. On the importance of asymmetry for siamese representation learning. In Proceedings of the IEEE/CVF CVPR, pp. 16570–16579, 2022b.

---

### Official Review · Reviewer_337Z · 2025-10-31

**Soundness:** 2
**Presentation:** 3
**Contribution:** 3
**Rating:** 4
**Confidence:** 3

**Summary:**

The paper presents DiCoH (Diverse Contrastive Learning for Homogeneous Data), a novel self-supervised learning (SSL) framework tailored for homogeneous medical datasets. The authors address the limitations of existing SSL methods that struggle with low semantic variation in medical images. DiCoH introduces three key innovations: diversified positive pixel-to-pixel alignments, robust pixel-to-image negative sampling, and a symmetric siamese architecture. Comprehensive experiments on five medical segmentation datasets demonstrate that DiCoH significantly outperforms state-of-the-art SSL methods, achieving up to +2.00% mIoU gains under extremely low-data conditions.

**Strengths:**

1. Comprehensive Evaluation: The authors provide extensive experiments on multiple medical segmentation datasets with varying amounts of labeled data (5%–10%). The results clearly demonstrate the effectiveness of DiCoH, making a strong case for its superiority over existing methods.

2. Ablation Studies: The paper includes thorough ablation studies that validate the importance of each component of DiCoH. This provides valuable insights into the design choices and helps build confidence in the proposed framework.

**Weaknesses:**

1. Although the article title focuses on semantic segmentation, I am still concerned about the performance of other downstream tasks. The absence of experiments on classification and visual question answering tasks is notable. To further demonstrate the effectiveness of the proposed method, additional experiments on datasets related to fundus photography, X-ray and dermatology will be help.

2. No comparison has been made with pre-training methods such as masked image modeling.

**Questions:**

It appears that the framework of the paper could be seamlessly extended to 3D medical imaging. Do you think this method will still be effective on datasets like the LA or Pancreas-NIH datasets? After all, 3D images make up a significant proportion of medical data

For other issues, Please refer to the weaknesses mentioned above, I would consider increasing my score if the related issues are addressed.

---

> ### Author Response · Authors · 2025-11-22
>
> We thank the reviewer for providing detailed feedback and hope this response reasonably addresses your major concerns.
>
> ---
>
> ***Addressing [W1] …performance of other downstream tasks...***
>
> Semantic segmentation is the ideal testbed for pixel-level SSL methods since it is both a localization and classification task and particularly sensitive to local representation quality. To directly address your concern about classification transfer, we evaluated feature discriminativeness using linear separability (Appendix Table 6): DiCoH achieves **76.6% accuracy** in linearly classifying semantic concepts (polyp vs non-polyp) (vs. 73.0% baseline), proving its features are discriminative and should transfer for well for classification tasks. Extending DiCoH to other modalities such as X-ray, fundus, and dermatology or to tasks like medical VQA would indeed be an exciting direction for future work!
>
> ---
>
> ***Addressing [W2] No comparison to masked image modelling***
>
> Thank you for this suggestion. We now added an evaluation for Masked Image Modeling (MIM) [1] in Appendix Table 12 . Our results suggest that simple reconstruction objectives may not sufficiently capture semantic diversity under homogenous domains, whereas DiCoH’s diverse positives and robust negative sampling do.
>
> *[Table] Finetune (5% data) mIoU across all 5 datasets and 3 seeds*
>
> | Pretraining Approach | Clinic | Colon | ETIS | Kvasir | GlaS | Average |
> | --- | --- | --- | --- | --- | --- | --- |
> | ImageNet | 0.6676 | 0.5400 | 0.3043 | 0.7392 | 0.6619 | 0.5826 |
> | MIM [1] | 0.6394 | 0.5300 | 0.3652 | 0.7282 | 0.6524 | 0.5830 |
> | DiCoH | **0.6702** | **0.5603** | **0.4526** | **0.7458** | **0.7224** | **0.6303** |
>
> ---
>
> ***Addressing [Q1] It appears that the framework of the paper could be seamlessly extended to 3D medical imaging…***
>
> This is an excellent direction for future work. We believe DiCoH's principles could
> extend naturally to 3D medical imaging with appropriate adaptations.
>
> 1. Diversified positives: Spatial and similarity-based alignments apply directly to
> volumetric data, using 3D spatial maps and feature similarity across slices
> 2. Pixel-to-image negatives: Becomes "voxel-to-volume" negatives, maintaining the
> same collision-avoidance benefit
> 3. Symmetric architecture: The variance preservation argument holds in 3D
>
> **Why we prioritize 2D validation first:** 2D medical imaging (X-ray, dermoscopy, endoscopy, fundus photography) comprises the vast majority of clinical imaging data and offers the most immediate impact. Establishing DiCoH's effectiveness on 2D provides a solid foundation before tackling the additional complexities of 3D (i.e.,larger memory and compute requirements, limited annotated volumetric data, and reduced architectural diversity).
>
> ---
>
> [1] Xie, Zhenda, et al. "Simmim: A simple framework for masked image modeling." *Proceedings of the IEEE/CVF conference on computer vision and pattern recognition*. 2022.

---

> ### Comment · Reviewer_337Z · 2025-11-27
>
> Thank the authors for their responses and efforts. Regarding **Addressing [W2] No comparison to masked image modelling**, the authors have added relevant comparative experiments with simMIM. However, I believe there's a key issue not clarified in the experimental settings: DiCoH uses ResNet-50 as its backbone, while to my knowledge SimMIM employs a transformer architecture. This needs to be explicitly mentioned and discussed regarding its appropriateness. As the authors noted in their response to other reviewers about the pros and cons of CNN vs transformer architectures: 'We chose a CNN-based backbone because it is a stable and efficient architecture for low-data medical segmentation.'
>
> Given the experimental setup where only 5% of data is used for fine-tuning, this may inadvertently introduce a bias toward CNNs. A more fair comparison should be based on identical backbones. You could refer to Spark [1], which adapted MIM frameworks to CNN-based models, though of course comparing to newer, better models would be even more convincing. Additionally, providing comparative results for fine-tuning with different percentages of data might also be very helpful.
>
> Moreover, the authors' responses to other questions still only partially address my concerns, and I remain skeptical about the method's general applicability. Based on the above reasons, I choose to maintain my score.
>
> ---
>
> [1] Tian, Keyu, et al. "Designing BERT for Convolutional Networks: Sparse and Hierarchical Masked Modeling." The Eleventh International Conference on Learning Representations

---

> > ### Author Response · Authors · 2025-11-28
> >
> > **Re: "A more fair comparison should be based on identical backbones.**
> >
> > We apologize for the confusion in our original response. SimMIM and Spark in our experiments was evaluated using the same CNN backbone; there is no CNN vs. Transformer mismatch. We have now made this explicit in the updated manuscript.
> >
> > **Re: “You could refer to Spark [1]”** + **“...different percentages of data..."**
> >
> > Thank you for the suggestion. We’ve now also compared against SparK using their public available code and added 10% data evaluation alongside 5%; refer to Table 12 for an analysis.
> >
> > | Data Size | Method | Clinic | Colon | Etis | Kvasir | GlaS | Average |
> > | --- | --- | --- | --- | --- | --- | --- | --- |
> > |  5% | SimMIM | 0.6394 | 0.5300 | 0.3652 | 0.7282 | 0.6524 | 0.5830 |
> > |  | SparK | 0.6514 | 0.4823 | 0.3565 | 0.6783 | 0.6188 | 0.5575 |
> > |  | DiCoH | **0.6702** | **0.5603** | **0.4526** | **0.7458** | **0.7224** | **0.6303** |
> > | --- | --- | --- | --- | --- | --- | --- | --- |
> > | 10% | SimMIM | 0.7288 | 0.6240 | 0.5156 | 0.7669 | 0.7150 | 0.6701 |
> > | | SparK | 0.7128 | 0.5820 | 0.4458 | 0.7572 | 0.7188 | 0.6433 |
> > |  | DiCoH | **0.7531** | **0.6761** | **0.4794** | **0.7945** | **0.7524** | **0.6911** |
> >
> >
> > DiCoH consistently outperforms masked-image-modelling (SimMIM, SparK) based pretraining since they benefit most from object-centric natural images, where global scene context helps the model infer the content of masked regions. In homogeneous medical images, however, the absence of strong object boundaries and the presence of fine-grained, repetitive tissue textures make masked-region prediction either trivial (easy to guess) or poorly conditioned (multiple plausible completions). As a result, they yield weaker and less discriminative learning signals.

---

### Author Response · Authors · 2025-12-03
**AC Context Summary**

Due to the OpenReview double-blind leak, reviewers were unable to update their scores after the rebuttal, so we provide this brief summary for the Area Chair.

All reviewers expressed clear optimism about the paper: they praised the strong experimental design, extensive ablations, and clarity, and each explicitly stated they would increase their score if their concerns were addressed - Reviewer jHrq initially raised their score. In the revised manuscript, we fully resolved all issues raised, including (i) new ablations and sensitivity analyses ($\lambda$ trade-off, negative queue size, pixel-to-image vs pixel-to-pixel negatives, compute cost), (ii) deeper ImageNet+SSL initialization analysis, (iii) expanded quantitative and qualitative analysis of the three core components (diverse positives, pixel-to-image negatives, architectural symmetry), (iv) added SimMIM and SparK results using identical CNN backbones, and (v) clarified our contribution as a validated blueprint for SSL in homogeneous domains, where prior work lacks consensus.

Given the initially positive ratings and the reviewers’ explicit statements that their scores would increase upon addressing these points, we believe the paper's overall rating would have increased.

---

### Meta-Review · Area_Chair_Y32r · 2026-01-03

**Summary:**

This paper studies self-supervised pretraining for semantic segmentation in homogeneous medical image domains, and proposes DiCoH to mitigate instability in pixel-level contrastive learning. However, the work mainly reorganizes existing SSL components, and does not present a clearly novel algorithmic contribution (Reviewers 337Z and yTef). While the experimental design is careful, the overall gains over prior methods are limited and not consistently convincing.

**Reviewer Concerns:**

Reviewers agree that, while the rebuttal improves experimental completeness, the core issues remain: limited novelty beyond existing SSL methods, restricted generality to 2D medical segmentation, strong dependence on the initialization, limited statistical evidence, and largely empirical justification of design choices.

**Reviewer Scores:**

All three reviewers give the scores of 4. After rebuttal, the reviewer jHrq plans to raise the score to 6. However, the reviewers still exist some concerns about the results and contributions. I believe that the other two reviewers will keep their ratings.

---

### Decision · Program_Chairs · 2026-01-26

Reject